# Is short-term-variation of fetal-heart-rate a better predictor of fetal acidaemia in labour? A feasibility study

Habiba Kapaya[1]*, Richard Jacques[2], Thomas Almond[3], Miss Hilary Rosser[3], Dilly Anumba[4]

**1** Sheffield Teaching Hospitals, NHS Foundation Trust, Tree Root Walk, Sheffield, United Kingdom, **2** Medical Statistics Group, School of Health and Related Research (ScHARR), University of Sheffield, United Kingdom, **3** Obstetrics and Gynaecology, Sheffield Teaching Hospitals, NHS Foundation Trust, Tree Root Walk, Sheffield, United Kingdom, **4** Academic Unit of Reproductive and Developmental Medicine, University of Sheffield, Tree Root Walk, Sheffield, United Kingdom

* h.kapaya@sheffield.ac.uk

**Data Availability Statement:** All relevant data are within the paper.

**Funding:** The paper presents independent research funded by the National Institute for Health

## Abstract

### Background

Continuous intrapartum fetal monitoring is challenging and its clinical benefits are debated. The project evaluated whether short-term-variation (STV) and other computerised fetal heart rate (FHR) parameters (baseline FHR, long-term-variation, accelerations and decelerations) predicted acidaemia at birth. The aims of the study were to assess the changes in FHR pattern during labour and determine the feasibility of undertaking a definitive trial by reporting the practicalities of using the monitoring device, participant recruitment, data collection and staff training.

### Methods

200 high-risk women carrying a term singleton, non-anomalous fetus, requiring continuous FHR monitoring in labour were consented to participate from the Jessop Wing maternity unit, Sheffield, UK. The trans-abdominal fetal ECG monitor was placed as per clinical protocol. During the monitoring session, clinicians were blinded to the computerised FHR parameters. We analysed the last hour of the FHR and its ability to predict umbilical arterial blood pH <7.20 using receiver operator characteristics (ROC) curves.

### Results

Of 200 women, 137 cases were excluded as either the monitor did not work from the onset of labour (n = 30), clinical staff did not return or used the monitor on another patient (n = 37), umbilical cord blood not obtained (n = 25), FHR data not recorded within an hour of birth (n = 34) and other reasons (n = 11). In 63 cases included in the final analysis, the computer-derived FHR parameters did not show significant correlation with umbilical artery cord pH <7.20. Labour was associated with a significant increase in short and long term variation of

Research (NIHR) under its Research for Patient Benefit (RfPB) Programme (Grant Reference Number PB-PG-1215-20010. The views expressed are those of the author(s) and not necessarily those of the NIHR or the Department of Health and Social Care.

**Competing interests:** The authors have declared that no competing interests exist.

**Abbreviations:** AUC, Area under curve; Bpm, beats per minute; CTG, Cardiotocography; CI, Confidence interval; EFM, Electronic fetal monitoring; FHR, Fetal heart rate; LTV, Long-term-variation; Ms, millisecond; PIS, Participant information sheet; ROC, Receiver operator curve; STV, Short-term-variation.

FHR and number of deceleration (P<0.001). However, baseline FHR decreased significantly before delivery (P<0.001).

## Conclusions

The project encountered a number of challenges, with learning points crucial to informing the design of a large study to evaluate the potential place of intrapartum computerised FHR parameters, using abdominal fetal ECG monitor before its clinical utility and more widespread adoption can be ascertained.

## Introduction

Reduced fetal oxygenation during labour remains an important cause of perinatal mortality and long-term neurologic morbidity [1]. Conventional cardiotocography (CTG) also called electronic fetal monitoring (EFM) is the most widespread method of fetal surveillance worldwide, and can detect fetuses exposed to decreased oxygen supply during labour, allowing early obstetric interventions. However, despite the widespread use of electronic fetal-heart-rate (FHR) monitoring, its potential for improving perinatal outcome has not been established. The reasons for this are complex, but include difficulty in interpreting the FHR traces during labour [2, 3].

Computerised analysis of CTG is a promising alternative to reduce poor reproducibility and overcome the unacceptably high intra- and inter-observer variability associated with the visual analysis of the CTG [4, 5]. The computerised CTG developed by Dawes and Redman enables measuring the short-term-variation (STV) of the FHR, which cannot be assessed by the naked eye [6, 7]. STV is one of the best predictors for fetal acid-base status before labour. However, its value in the intrapartum period has not been realised.

The aims of our study were to: 1) assess the diagnostic accuracy of STV for predicting fetal acidaemia at birth; 2) assess whether other computerised FHR parameters predict fetal acidaemia at birth; 3) study changes in computerised FHR parameters in relation to labour events that may contribute to fetal acidaemia; 4) evaluate the feasibility of undertaking a large multicentre study.

## Material and methods

### Participants and setting

This was a single-centre prospective study conducted on 200 women with risk factors qualifying for continuous CTG in labour, at the Jessop Wing Hospital in Sheffield, UK between January 2018 and May 2019. The research protocol was approved by the Fulham Research Ethics committee, London (17/LO/1236).

Women were eligible for participation in the study if they fulfilled the following inclusion criteria: singleton pregnancy with a cephalic presentation, $\geq$ 36 completed weeks of gestation, no known major fetal malformation, in active labour but not in second stage, no known contraindication to vaginal delivery, a clinical decision had been made for continuous FHR monitoring in labour, $\geq$ 18 years of age and able to provide written informed consent.

### Trans-abdominal fetal ECG monitor (Monica AN24)

The FHR recordings were performed with the trans-abdominal ambulatory fetal ECG monitor (Monica AN24, Monica Healthcare Nottingham, UK). The device is attached via a cable

assembly which in turn attaches to five standard disposable electrodes placed on the abdomen of a pregnant woman [8]. The methodology used for signal extraction and analysis has been described in detail by Cohen et al. [9]. Although the fetal ECG monitor uses different signals to acquire FHR data, the monitor relies on the similar algorithm as introduced by Dawes and Redman to calculate STV and other automated FHR parameters [10]. This is in contrast to the ultrasound-based CTG monitors.

The FHR parameters studied from the Dawes Redman analysis for this study were:

- Basal FHR: baseline heart rate measured in beats per minute (bpm).

- Long-term FHR variation (LTV): minute-by-minute range of pulse intervals.

- Short-term FHR variation (STV): sequential epoch-to-epoch variation. Both the LTV overall and the STV overall are measured in milliseconds (ms).

- Accelerations: increase in FHR above the baseline that lasted longer than 15 s and had an amplitude greater than 10 bpm.

- Decelerations: decrease in FHR below the baseline that lasted longer than 15 s and had an amplitude greater than 10 bpm.

### Recruitment and procedure

The fetal ECG monitor was fitted on the maternal abdomen during labour. During the monitoring session, clinicians were blinded to the STV and additional computerised FHR parameters. However, conventional FHR patterns obtained from the Monica AN24 were displayed on the monitor and were employed for the intrapartum clinical care of women enrolled for the study. The midwife who cared for the women in labour was advised to use the "event" button on the monitor to record events during labour, such as the administration of medicine that has been shown to influence FHR patterns including STV [11]. At the end of the monitoring session the device was returned by the clinical midwife to a designated place and collected by a research midwife. The intrapartum FHR data were downloaded and stored on a password protected computer. The STV and other computerised FHR parameters were studied at the completion of the study. To evaluate whether STV and other computerised FHR were related to fetal acidaemia and the associated outcomes of the infants, we analysed the last 60 minutes of this data before birth.

### Sample size

Based on 2015–2016 birth statistics at the Jessop Wing Hospital Sheffield, 10% prevalence of acidaemia (umbilical artery cord pH<7.20) was observed in women requiring continuous intrapartum fetal monitoring. With the sample size of 200 participants and the prevalence of acidaemia of 10% (the event rate) in our population, we expected 20 births to be affected by a degree of acidaemia. The sample size was calculated to estimate the predictive potential for fetal acidaemia: with 200 participants and a prevalence of 10%, the area under the ROC curve of 0.8 could be estimated with a precision half width of a 95% confidence interval) of 0.12 [12].

### Outcome measure

The umbilical cord pH is considered a crucial outcome measure and one of the essential criteria for the diagnosis of acute intrapartum hypoxic events [13, 14]. Neonatal acidaemia based on arterial cord blood analysis has traditionally been defined as pH <7.20 [15]; however other studies suggest pH <7.00 to be associated with major neurological morbidity [16]. At present

the pH level constituting clinically significant acidaemia remains unclear [13]. Nevertheless, it is generally agreed that the risk of neonatal complication increases as the pH at birth decreases [17]. For the purpose of this study, an umbilical arterial blood pH threshold of <7.20 was used to discriminate acidaemic from non-acidaemic babies. Blood was taken from a double-clamped cord into a heparinised syringe and measured within 30 minutes of birth.

### Statistical analysis

The diagnostic accuracy of STV for predicting arterial cord pH < 7.20 was assessed by calculating the area under the ROC curve along with the associated 95% confidence interval (CI). This analysis was repeated for the other computerised FHR (baseline FHR, LTV, accelerations and decelerations). In addition, Mann-Whitney U tests were used to compare FHR parameters between those with cord pH < 7.20 and those with cord pH ≥ 7.20.

The final analysis compared FHR parameters between different events during labour. For baseline FHR, STV and LTV, linear mixed effects models were used, while for the count variables of accelerations and decelerations, Poisson mixed effects models were used. In all cases, a fixed factor for labour event was included to estimate the effect of each event compared to the reference event of first measurement when the Monica was put on,and a random intercept for participant was included to allow for there being multiple measurements per individual. A second model was fit for STV and LTV with an additional fixed factor included for cord pH group. The interaction between event and cord pH group was then tested to investigate if any differences between events differed by cord pH group. The estimates from the linear mixed effects models are presented as the difference in mean and the estimates from the Poisson mixed effects models as incidence rate ratios (IRR).

Statistical analysis was conducted in R version 3.6.3 [18]. ROC analysis was conducted using the pROC library [19], linear mixed effect models using the nlme library [20] and Poisson mixed effects models using the lme4 library [21].

## Results

Between January 2018 and May 2019, 200 women were recruited for the study. However, only 63 women were included in the final analysis as they had both arterial cord pH and STV measurement within 1 hour of delivery. The range of cord blood pH was from 6.997 to 7.416. The prevalence of cord blood pH < 7.2 was 33.3% (21/63), 95% CI: 22.9%-45.6% with 1.6% (1/63) having a pH less than 7.0 and 31.7% (20/63) having a pH less than 7.2 but greater than or equal to 7.1.

The flow diagram (Fig 1) explains recruitment and reasons for exclusion from the analysis. Clinical and demographic characteristics of those included in the study are shown in Table 1.

Comparison of STV and other computerised FHR parameters between those with umbilical artery cord pH<7.2 and those with pH≥7.2 showed no significant difference between the two groups for STV (P = 0.732), LTV (P = 0.789), baseline FHR (P = 0.240), accelerations (P = 0.549) and decelerations (P = 0.160); see Table 2.

The ROC analysis in Fig 2 showed that area under the ROC curve for STV predicting acidosis was 0.527 (95% CI: 0.377–0.678), suggesting that in this cohort STV is not a good predictor of acidaemia. The CI is wide due to the small sample size and the upper limit is below the 0.8 minimum value. Among other FHR parameters, decelerations had the largest point estimate of area under the curve (AUC = 0.609, 95% CI: 0.465 to 0.752) but this does not suggest that this is a good predictor of acidaemia. Like STV, the CI is wide due to the small sample size but the upper limit is close to the 0.8 minimum value as expected for this to be a useful diagnostic test.

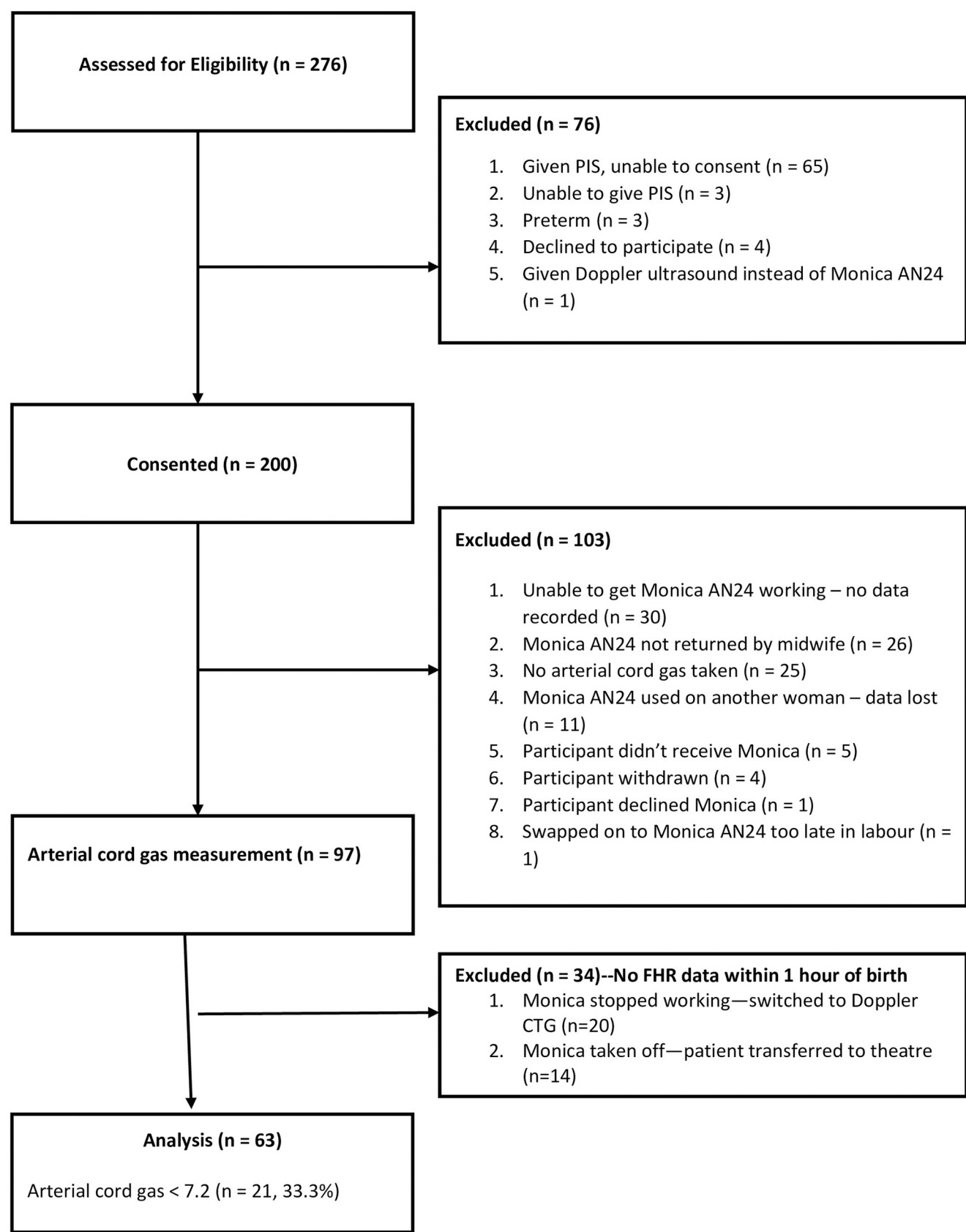

**Fig 1. Flow diagram showing the numbers of patients approached, excluded, recruited and included in data analysis.**

**Table 1. Summary statistics of study participants.**

| | Cord pH < 7.2 (n = 21) | Cord pH ≥ 7.2 (n = 42) | Overall (n = 63) | P-Value |
|---|---|---|---|---|
| Maternal Age (years), Median (IQR)[1] | 28.0 (25.0 to 32.0) | 30.5 (27.0 to 34.8) | 29.0 (26.5 to 34.0) | 0.113 |
| BMI (kg/m$^2$), Median (IQR)[1] | 27.5 (23.9 to 30.8) | 26.4 (22.9 to 32.4) | 26.4 (23.1 to 32.4) | 0.936 |
| Smoking Status, N (%)[2] | | | | |
| Yes | 3 (14.3%) | 6 (14.3%) | 9 (14.3%) | 1.000 |
| No | 18 (85.7%) | 36 (85.7%) | 54 (85.7%) | |
| Gestational Age (weeks), Median (IQR)[1] | 40.0 (39.0 to 41.0) | 40.0 (38.3 to 41.0) | 40.0 (39.0 to 41.0) | 0.210 |
| Birth Weight (g), Median (IQR)[1] | 3780 (3400 to 3940) | 3600 (3343 to 3755) | 3625 (3345 to 3835) | 0.229 |
| Onset of Labour, N (%)[2] | | | | |
| Spontaneous | 4 (19.0%) | 8 (19.0%) | 12 (19.0%) | 1.000 |
| Induction | 17 (81.0%) | 34 (81.0%) | 51 (81.0%) | |
| Mode of Delivery, N (%)[2] | | | | |
| Unassisted | 15 (71.4%) | 27 (64.3%) | 42 (66.7%) | 0.923 |
| Instrumental | 4 (19.0%) | 9 (21.4%) | 13 (20.6%) | |
| Caesarean | 2 (9.5%) | 6 (14.3%) | 8 (12.7%) | |
| Duration of Recording (min), Median (IQR)[1] | 533 (454 to 632) | 426 (247 to 592) | 485 (257 to 605) | 0.187 |
| Cord pH, Median (IQR) | 7.14 (7.12 to 7.15) | 7.25 (7.23 to 7.30) | 7.23 (7.16 to 7.27) | - |
| APGAR Score at 1 min, N (%)[2] | | | | 1.000 |
| <5 | 2 (9.5%) | 3 (7.1%) | 5 (7.9%) | |
| ≥5 | 19 (90.5%) | 39 (92.9%) | 58 (92.1%) | |
| APGAR Score at 5 min, N (%)[2] | | | | 0.333 |
| <7 | 1 (4.8%) | 0 (0.0%) | 1 (1.6%) | |
| ≥7 | 20 (95.2%) | 42 (100.0%) | 62 (98.4%) | |
| Admission to Neonatal Unit, N (%)[2] | | | | 1.000 |
| Yes | 1 (4.8%) | 2 (4.8%) | 3 (4.8%) | |
| No | 20 (95.2%) | 40 (95.2%) | 60 (95.2%) | |

[1] Between group comparison using Mann-Whitney U test.

[2] Between group comparison using exact test.

Fig 3A–3E compares the baseline FHR, STV, LTV, accelerations and decelerations between the first measurement when the fetal ECG monitor was put on and other labour events. The STV (2.35, 95% CI: 1.47–3.24, P<0.001), LTV (15.2, 95% CI: 10.4–20.0, P<0.001) and number

**Table 2. Comparing Monica parameters between outcome groups.**

| | Summary Statistic | Cord pH < 7.2 (n = 21) | Cord pH ≥ 7.2 (n = 42) | P-Value[1] |
|---|---|---|---|---|
| **Basal (bpm)** | **Median (IQR)** | 139.0 (121.0 to 147.0) | 128.5 (120.3 to 138.0) | 0.240 |
| | **Range** | 101.0 to 167.0 | 102.0 to 137.0 | |
| **STV (ms)** | **Median (IQR)** | 12.2 (10.3 to 14.2) | 12.0 (9.8 to 14.7) | 0.732 |
| | **Range** | 4.4 to 21.2 | 5.2 to 24.3 | |
| **LTV (ms)** | **Median (IQR)** | 65.6 (54.2 to 78.2) | 63.5 (53.9 to 79.9) | 0.789 |
| | **Range** | 24.5 to 124.6 | 53.9 to 79.8 | |
| **Accelerations** | **Median (IQR)** | 10 (7 to 14) | 9 (6 to 13) | 0.549 |
| | **Range** | 2 to 24 | 1 to 21 | |
| **Decelerations** | **Median (IQR)** | 4 (2 to 6) | 4 (1 to 5) | 0.160 |
| | **Range** | 1 to 15 | 0 to 15 | |

[1] Between group comparison using Mann-Whitney U test.

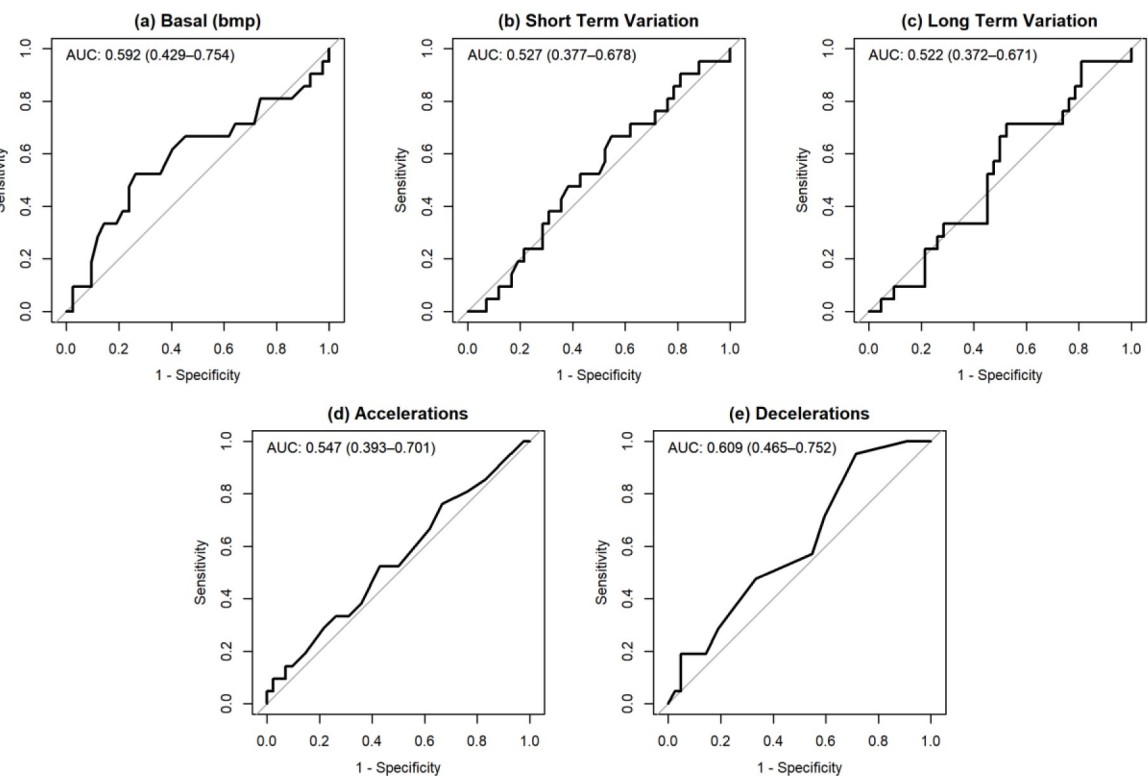

**Fig 2. ROC Curves for predicting umbilical artery cord pH with AUC (95% CI).**

of decelerations increased significantly before delivery (IRR: 2.75, 95% CI: 2.23–3.39, P<0.001), whereas the baseline FHR (-4.7, 95% CI: -7.6- -1.9, P = 0.001) decreased significantly as labour progressed.

We observed a significant reduction in baseline FHR and number of accelerations when epidural was sited (-5.0, 95% CI: -8.3- -1.7, P = 0.003; IRR: 0.66, 95% CI: 0.57–0.75, P<0.001) and diamorphine was administered (-5.7, 95% CI: -10.9- -0.6, P = 0.030; IRR: 0.53, 95% CI: 0.42–0.68, P<0.001). Number of accelerations also showed a significant drop when antihypertensives were administered (IRR: 0.56, 95% CI: 0.34–0.92, P = 0.021).

We found no statistically significant interactions between labour event and cord pH group for STV (P = 0.924) and LTV (P = 0.943). This suggests that any differences between labour events did not differ by cord pH.

## Discussion

Whilst the intention of this study was to explore the predictive utility of intrapartum STV and other computerised FHR parameters for diagnosing acidaemia at birth, the study also hoped to address important feasibility issues including, practicalities of using the monitoring device, participant recruitment and data collection.

We acknowledge several limitations of this study as the project encountered a number of challenges throughout with the learning points crucial to informing the design of any future multi-centre research.

In 50 participants, the device did not record the FHR data either from the beginning (n = 30) or during the last hour of delivery (n = 20). Our previous works [22–24] and that of other investigators [8, 9] have demonstrated the feasibility and accuracy of computerised FHR

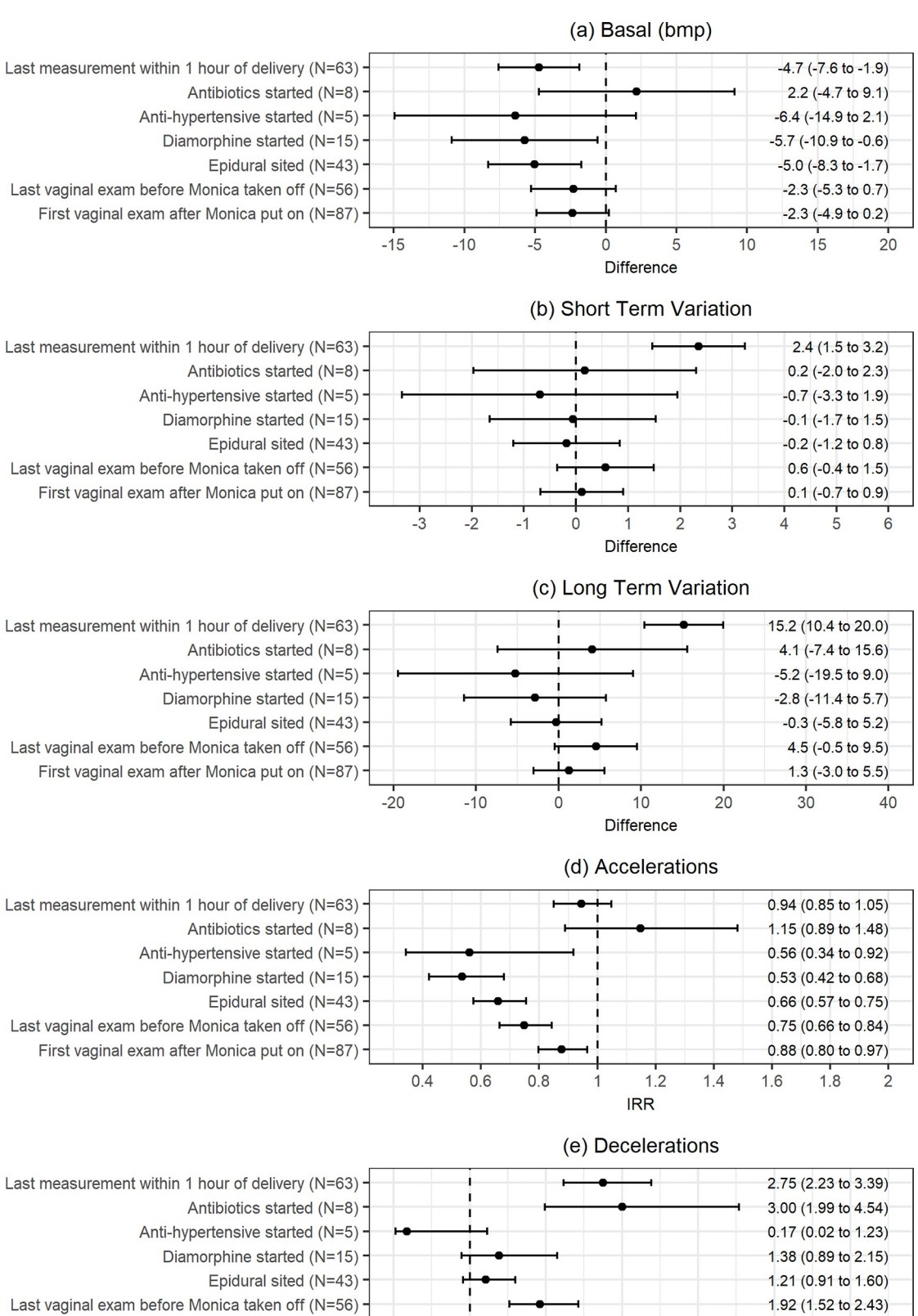

**Fig 3. Confidence interval plots comparing first measurement when fetal monitoring commenced to other labour events.**

monitoring using Monica AN24 during pregnancy. However, the benefits of computerised FHR pattern using Monica AN24 during labour remains unclear.

For the first time, a recently reported secondary analysis [25] from the previously published parent study [9] assessed whether Monica AN24 produced FHR patterns that could be interpreted and compared to those obtained by Doppler ultrasound. The parent study [9] was undertaken on 75 labouring women at three centres and compared the two modalities of monitoring (abdominal fetal ECG and Doppler ultrasound) with fetal scalp electrode. The study [9] showed that FHR detection using abdominal fetal ECG device was more reliable and accurate than Doppler ultrasound. For the purpose of secondary analysis [25], the authors reported data on the subset of women (n = 30) instead of including all 75 women that participated in the parent study [9]. Results of this secondary analysis [25] showed that compared to Doppler ultrasound, abdominal fetal ECG reliably reproduced the computerised FHR patterns during labour. However, this finding should be interpreted with caution as it was from a portion of data from a previously published parent study [9] that was not powered or designed to address the specific hypothesis of this secondary analysis.

The other issue that we encountered during this period was related to human factors. In 37 cases, the recorded FHR data was lost as the monitor was used on a new patient prior downloading the FHR data from the recruited women. As this problem became more apparent, the research midwives worked with the midwifery staff on the labour ward to prevent as much data loss as possible. Simplified data collection forms were provided with the monitors so the appropriate information was recorded in real time. The data collection sheet also gave clear instructions on secure returning of the device. In addition, education sessions were utilised for training new midwives coming for rotations on the labour ward to ensure they had better understanding of the research project.

We excluded 25 women from the final analysis as there was no record of umbilical cord blood results. In our institution, cord blood sampling is not routinely performed as a means of additional assessment in babies who are born in good condition. Although, women recruited for the study agreed to have cord blood taken after delivery, we suspect the clinical team forgot to obtain this as there was no clinical indication for this.

With recruitment nearing completion, the research team realised that the clinical midwives swapped the fetal ECG monitor with Doppler ultrasound when the clinical decision was made for delivering in theatre. Due to this approach, 14 patients were excluded from the final analysis as the downloaded data had no recording of STV or other computerised FHR pattern within an hour of delivery.

In view of the above issues, of the 200 women consented for the study, 137 were excluded from the final analysis. A significant loss of data was a massive blow to the project. However, strikingly, of the 63 women included in the final analysis, 21 delivered babies with an umbilical artery cord pH <7.20. The fact that we saw a higher prevalence of fetal aciademia in our sample mean that the prevalence used in the sample size was an underestimate of the true prevalence or it could be that this sample was not representative of 2015–2016 audit, implying that our study may have suffered selection bias.

During labour, the fetal circulation system changes because of placental insufficiency during contractions and fetal vagal nerve activation with a decrease in in the oxygen concentration of the blood [26]. Therefore, the decrease of baseline FHR together with a decrease in the complex activity of the fetal central nervous system reflects a physiological response to labour progression. We observed a similar response in our fetuses with a rise in number of decelerations and a fall in baseline FHR during second stage of labour when stressful stimuli are maximal.

Although neonatal outcome is generally, better with epidural analgesia than with parenteral opioid [27], both have the potential to alter FHR pattern [28, 29] making interpretation of fetal

CTG recordings potentially problematic. In our cohort, we observed a significant fall in baseline FHR and number of accelerations when epidural and diamorphine were administered. In addition, the number of accelerations were significantly affected when the women received antihypertensive treatment. The altered CTG pattern observed with the administration of an antihypertensive may be attributable to underlying placental insufficiency that can be associated with hypertensive disorder or the effect of anti-hypertensives that may alter maternal hemodynamics [30, 31].

We analysed end of labour FHR data which are fundamentally different from pre-labour (antepartum) FHR data [5]. Antepartum FHR data are more likely to be stable and usually do not have decelerations, whereas end-of-labour traces are more likely to be unstable, with frequent signal loss and FHR decelerations [5, 32] as observed in our study.

It is known that reduced STV in antepartum traces is an ominous sign [5, 6]. Our work shows that STV value increase at birth. This inverse relationship has been shown in a study where an acute response to experimental repeated asphyxia in the term fetus resulted in an increase in FHR variability instead of a decrease [33].

In summary, although Monica AN24 is truly non-invasive and does not expose the fetus to ultrasound waves, proving the clinical utility of transabdominal fetal ECG is a prerequisite to its wider adoption. The main motivation for using transabdominal fetal ECG device remains its superiority for monitoring obese women rather than potential signal-analytical benefits of high quality fetal ECG signal [34, 35]. In addition, this technology is available in few delivery centres with a limited number of devices [34].

Our work indicates technical issues related to the monitoring device as well as challenges that were associated with the training and engagement of clinical midwives resulting in a significant barrier to recruitment as well as loss of significant clinical data. However, despite these limitations, further progress is possible. We advocate future development of innovative fetal surveillance monitors and improvement in the signal processing methods of trans-abdominal fetal ECG technology and appropriate data collection. This would allow objective and standardised reporting of diagnostic accuracies of computerised FHR parameters. We envisage that technological progress will happen through better quality and more reliable acquisition of FHR data and through novel techniques to acquire additional information about the fetus during labour. For example, to complement analysis of FHR variability, morphological analysis of the abdominal fetal ECG could potentially play a crucial role. This remains to be investigated and future research can address this pertinent question. Finally, education and engagement activities for clinical staff to recognise the power of clinical data, employing computers to help understand the relations between FHR and neonatal outcome is crucial when designing and planning future work.

## Conclusion

In view of the high attrition rate and small number of cases analysed, valid conclusions could not be drawn from this study. However, we believe that in view of the importance of consequences of intrapartum hypoxia, progress is required by undertaking multidisciplinary collaborative research. This can be achieved through carefully designed clinical studies of the new and existing fetal monitoring techniques with careful consideration given to learning from mistakes from the preliminary work.

## Acknowledgments

The Yorkshire and Humber Clinical Research Network provided additional support to the study. We would like to thank Mr James Bushby for the technical support in kind for the study and Professor Tracey Young for reading the final version of the manuscript.

## Author Contributions

**Conceptualization:** Habiba Kapaya.

**Data curation:** Thomas Almond, Miss Hilary Rosser.

**Formal analysis:** Richard Jacques.

**Funding acquisition:** Habiba Kapaya.

**Methodology:** Habiba Kapaya, Thomas Almond.

**Project administration:** Habiba Kapaya, Thomas Almond.

**Supervision:** Dilly Anumba.

**Writing – original draft:** Habiba Kapaya, Richard Jacques.

**Writing – review & editing:** Habiba Kapaya, Dilly Anumba.

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
