## [Decision Letter · Decision Letter 0]

11 Jun 2020

PONE-D-20-14044

"Is short-term-variation of fetal-heart-rate a better predictor of fetal acidaemia in labour? A feasibility study"

PLOS ONE

Dear Dr. Kapaya,

Thank you for submitting your manuscript to PLOS ONE. After careful consideration, we feel that it has merit but does not fully meet PLOS ONE’s publication criteria as it currently stands. Therefore, we invite you to submit a revised version of the manuscript that addresses the points raised during the review process.

SPECIFIC ACADEMIC EDITOR COMMENTS: There were two experts in the field that handled your manuscript. We thank them for their time. Although they found interest in your study, several major comments arose during review. These comments include, but are not limited to: the need for clarification of vague statements and concerns related to the statistical analysis. Please address all comments.

We look forward to receiving your revised manuscript.

Kind regards,

Frank T. Spradley

Academic Editor

PLOS ONE

Journal Requirements:

'The study was funded by the NIHR, Research for Benefit (RfPB), programme; grant reference PB-PG-1215-20010. Additional support was provided from the Yorkshire and Humber Clinical Research Network'

'Dr Habiba Kapaya was awarded £116,333 by the NIHR Research for Patient Benefit (RfPB) programme to undertake this study'

Reviewers' comments:

Reviewer's Responses to Questions

**Comments to the Author**

1. Is the manuscript technically sound, and do the data support the conclusions?

Reviewer #1: Yes

Reviewer #2: Yes

2. Has the statistical analysis been performed appropriately and rigorously? 

Reviewer #1: Yes

Reviewer #2: I Don't Know

3. Have the authors made all data underlying the findings in their manuscript fully available?

Reviewer #1: Yes

Reviewer #2: Yes

4. Is the manuscript presented in an intelligible fashion and written in standard English?

Reviewer #1: Yes

Reviewer #2: No

5. Review Comments to the Author

Reviewer #1: This is a well-organized and clearly written manuscript. The problem, also well described by the authors, is the large attrition in the study population and consequently the very limited number of cases with acidemia most of which is mild and thus unlikely to have an association with adverse outcome.

That said, the results are of interest and contribute to a growing body of evidence that STV does not predict acidemia. This is an important finding because many clinical guidance are based on the assumption that metabolic acidemia does not exist when heart rate variability is within the normal range.

Comments:

1. The definition of acidemia is generous. A pH of 7.20 is not associated with adverse outcomes In fact in one very large Oxford -based study of gases, the mean umbilical artery pH at birth was 7.208. (Eur J Obstet Gynecol Reprod Biol 2013 Jun;168(2):155-60.) Many others report mean UA pH values around 7.24 to 7.28. The reported IQR in this study indicates that there were probably only 5 cases with pH<7.12. Please provide the number and percentage of cases with pH< 7.10 and pH <7.0.

2. Please temper your results to indicate that STV as measured by the Dawes algorithm over the last 60 minutes of tracing before birth did not discriminate the control group from the acidemia group as they were defined in this study where most of the acidemia was between 7.12-7.15.

3. This study found that in general STV and LTV increased over time. (Figure 3) I believe this is an analysis of all cases. This is an interesting finding, but the general objective of this study is to determine if the two subgroups were different. Please compare the two study subgroups regarding their change in STV and LTV.

4. Lines 193-194. This sentence is vague. What do you mean? Is the reliability and accuracy of the Dawes algorithm unmeasured? or unsatisfactory? Is the reliability and accuracy of the combination of the Dawes algorithm using Monica data unmeasured? or unsatisfactory? The Dawes algorithm and the Monica device are the basic “instruments” of your study. Do you lack confidence in the accuracy of this combination to measure variability? Please clarify.

5. Consider for example, it is possible that the STV increases with mild acidemia and then falls with more extreme aberrations. Such a speculation cannot be examined without adequate representation of cases along the entire spectrum of pH measurements. Such a bimodal response would make comparison of means or ROC curves difficult to interpret because both very high and very low variability may be a predictor of acidemia. Confine your conclusions to precisely what your data can support.

In short, while this is a very small study, it does have value. Conclusions must be stately in a precise manner because the study sample is very limited in terms of size and degree of clinically meaningful abnormality.

Reviewer #2: This research manuscript presents the findings with respect to intrapartum fetal heart rate (FHR) monitoring using the transabdominal ambulatory fetal ECG monitor (Monica AN24). This technology is not in wide use, particularly intrapartum , but the authors present its potential value and review previous experience. In this manuscript they state three aims exploring its value in providing computerized FHR analysis of short term variation(STV) in particular, but also other FHR parameters. A fourth aim was to assess feasibility of their protocol for a larger, possibly multicenter study.

Below my comments will be divided under the authors' headings, sometimes indicating Line Number.

Sample Size: A reference should be provided for the sample size calculation using receiver operating characteristic curve (ROC) area under the curve (AUC) as the basis. Most Ob/Gyn readers, including this reviewer, are not familiar with this approach to sample size calculation. Presumably 'precision' is 95% confidence interval (CI). Please clarify.

Statistical Analysis: More detail on the manner of the analysis (regression ?) should be included. The statistical computer package used for the regression, and non-parametric analyses, but particularly the ROC analysis, should be stated.

Results: In Figure 1, the flow diagram, the abbreviation 'PIS' is used and not defined.

Discussion:

Line 240: ? typo 'declarations' should be 'decelerations'

Line 262: 'superiority for monitoring obese women'

There is no data in this study to support this statement. If this is based on prior research, a reference should be provided. If true, this would be a very attractive feature.

References: The citations for reference #20 and #27 are incomplete.

My answer to PLOS Question 4,'No', is because of the minor typos and omissions just identified, which are easily corrected.

The authors are correct. Valid conclusions could not be drawn from their data. However they provide details of the many pitfalls they encountered with the intrapartum use of this FHR technology, and their study protocol, which will be useful for future studies to consider in design, and hopefully avoid or minimize. As such, with revision, I support acceptance for publication.

6. PLOS authors have the option to publish the peer review history of their article (what does this mean?). If published, this will include your full peer review and any attached files.

Reviewer #1: No

Reviewer #2: Yes: David C. Young, Dalhousie University and IWK Health Centre, Halifax, NS, Canada

---

## [Author Response · Author response to Decision Letter 0]

3 Jul 2020

A file labelled "Response to Reviewers' comments" is uploaded. This document responds to each point raised by the academic editor and reviewers.

---

## [Decision Letter · Decision Letter 1]

20 Jul 2020

"Is short-term-variation of fetal-heart-rate a better predictor of fetal acidaemia in labour? A feasibility study"

PONE-D-20-14044R1

Dear Dr. Kapaya,

We’re pleased to inform you that your manuscript has been judged scientifically suitable for publication and will be formally accepted for publication once it meets all outstanding technical requirements.

Kind regards,

Frank T. Spradley

Academic Editor

PLOS ONE

Reviewers' comments:

Reviewer's Responses to Questions

**Comments to the Author**

1. If the authors have adequately addressed your comments raised in a previous round of review and you feel that this manuscript is now acceptable for publication, you may indicate that here to bypass the “Comments to the Author” section, enter your conflict of interest statement in the “Confidential to Editor” section, and submit your "Accept" recommendation.

Reviewer #1: All comments have been addressed

Reviewer #2: All comments have been addressed

2. Is the manuscript technically sound, and do the data support the conclusions?

Reviewer #1: Yes

Reviewer #2: (No Response)

3. Has the statistical analysis been performed appropriately and rigorously? 

Reviewer #1: Yes

Reviewer #2: (No Response)

4. Have the authors made all data underlying the findings in their manuscript fully available?

Reviewer #1: Yes

Reviewer #2: (No Response)

5. Is the manuscript presented in an intelligible fashion and written in standard English?

Reviewer #1: Yes

Reviewer #2: (No Response)

6. Review Comments to the Author

Reviewer #1: (No Response)

Reviewer #2: (No Response)

7. PLOS authors have the option to publish the peer review history of their article (what does this mean?). If published, this will include your full peer review and any attached files.

Reviewer #1: No

Reviewer #2: **Yes: **David C. Young MD, FRCSC, Department of Obstetrics and Gynecology, Dalhousie University and IWK Health Centre, Halifax, NS, Canada

---

## [Editor Report · Acceptance letter]

22 Jul 2020

PONE-D-20-14044R1 

"Is short-term-variation of fetal-heart-rate a better predictor of fetal acidaemia in labour? A feasibility study" 

Dear Dr. Kapaya:

I'm pleased to inform you that your manuscript has been deemed suitable for publication in PLOS ONE. Congratulations! Your manuscript is now with our production department. 

Kind regards, 

on behalf of

Dr. Frank T. Spradley 

Academic Editor

PLOS ONE